

**Profiling of CH4 background mixing ratio in the lower troposphere with Raman lidar: a feasibility experiment.**

Igor Veselovskii[1], Philippe Goloub[2], Qiaoyun Hu[2], Thierry Podvin[2], David .N. Whiteman[3], Mikhael Korenskiy[1], Eduardo Landulfo[4]

[1]*Physics Instrumentation Center of General Physics Institute, Troitsk, Moscow, Russia.*

[2]*Laboratoire d'Optique Atmosphérie, Université de Lille-CNRS, Villeneuve d'Ascq, France*

[3]*NASA Goddard Space Flight Center, Greenbelt, USA*

[4]*Instituto de Pesquisas Energeticas e Nucleares, Sao Paulo, Brazil*

**Abstract**

We present the results of methane profiling in the lower troposphere using LILAS Raman lidar from Lille University observatory platform (France). The lidar is based on a tripled Nd:YAG laser and nighttime profiling up to 4000 m with 100 m height resolution is possible for methane. Agreement between measured the photon counting rate in the $CH_4$ Raman channel in
the free troposphere and numerical simulations for a typical $CH_4$ background mixing ratio (2 ppm) confirms that $CH_4$ Raman scattering is observed. Within the planetary boundary layer, an increase of the $CH_4$ mixing ratio, up to a factor of 2, is observed. Different possible interfering factors, such as leakage of the elastic signal and aerosol fluorescence have been taken into consideration. Tests using backscattering from clouds confirmed that the filters in the Raman
channel provide sufficient rejection of elastic scattering. The measured methane profiles do not correlate with aerosol backscattering, which corroborates the hypothesis that, in the PBL, not aerosol fluorescence but $CH_4$ is observed. However, the fluorescence contribution cannot be completely excluded and, for future measurements, we plan to install an additional control channel close to 393 nm where no strong Raman lines exist and only fluorescence can be
observed.

## 1.    Introduction.

Raman spectroscopy is a powerful technique for identification of different gases in the atmosphere and for the estimation of their concentration (Weber, 1979), which can be used in
conjunction with lidar technology (Inaba and Kobayasi, 1972). An example of such synergy is



the Raman lidar for water vapor monitoring (Whiteman et al., 1992). For optimum application of the Raman technique, the gas of interest should be abundant in the atmosphere, possess a large scattering cross section and have a Raman spectrum that is isolated from potential interfering species. Detection of water vapor with Raman spectroscopy satisfies all of these conditions and

has become a very popular application of lidar (e.g. Whiteman et al, 2007 and references therein). Besides water vapor, Raman lidar profiling of carbon dioxide (Ansmann et al., 1998; Whiteman et al., 2007; Zhao et al., 2008) as well as quartz crystals in dust layers (Müller et al., 2010) has been reported.

Methane is currently the second (after carbon dioxide) most important greenhouse gas of

anthropogenic origin (IPCC, 2013). Methane is emitted from a variety of natural and anthropogenic sources (e.g. Baray et al., 2018; Kavitha et al., 2016 and references therein) and on a per-molecule basis, methane is about 30 times more effective a greenhouse gas than carbon dioxide (Etminan et al., 2016). Global information about the $CH_4$ column concentration is available from satellite observations with, for example, the SCIAMACHY sensor on board

ENVISAT (Bovensmann et al., 1999) or the TANSO-FTS sensor on board GOSAT (Kuze et al., 2009). Today, it is well established that, in the free troposphere (FT) the $CH_4$ mixing ratio is about 2 ppm, while inside the planetary boundary layer (PBL), the mixing ratio can be significantly increased in the vicinity of methane sources (Baray et al., 2018). Such enhancement up to 4 ppm was observed, for example, in the airborne measurements over oil sands (Baray et

al., 2018). At low altitudes, the methane concentration depends on the PBL dynamics, so it is important to profile the methane mixing ratio simultaneously with the PBL parameters such as PBL height and aerosol backscattering. Profiling of the PBL is commonly done by aerosol lidars (Kovalev and Eichinger, 2004), while for methane profiling either the differential absorption (DIAL) or Raman lidar can be used.

Existing DIAL systems for measuring methane are based on tunable parametric laser sources and operate in the shortwave infrared (SWIR) spectral range between 1.65 and 2.3 μm, where methane has strong absorption lines (Refaat et al., 2013; Riris et al., 2017). Due to low Raleigh scattering in the SWIR region methane profiling is possible using the DIAL technique only inside regions containing significant amounts of aerosol. Raman lidars, by contrast, use

standard off-the-shelf tripled Nd:YAG lasers and are relatively simple in design. The methane molecule is quite suitable for Raman detection. The vibrational Raman line at 2914 cm$^{-1}$ is well





isolated and has the scattering cross section about eight times higher than that of nitrogen (Weber, 1979). The main difficulties of $CH_4$ Raman detection are related to its low background atmospheric concentration. The first attempts to implement $CH_4$ Raman spectroscopy in lidars go

back to eighties. Raman lidar was used for monitoring of methane plumes with relative $CH_4$ volume concentration of about 2% (Houston et al., 1986). Monitoring of the background $CH_4$ concentrations in the troposphere with airborne Raman lidar was reported by Heaps and Burris (1996). In both cases powerful excimer lasers (XeCl and XeF, respectively) were used. However, the wideband radiation of excimer lasers requires the use of wideband interference filters in

Raman channel, which, in turn, increases the sky background noise and possible contribution of aerosol fluorescence. Wideband detection also creates an additional complication related to interference from the oxygen Raman overtone (second Stokes shift) (Heaps and Burris, 1996). Significant progress in the development of the interference filters, detectors and laser sources during the last two decades provides, now the opportunity to develop the $CH_4$ Raman lidar based

on a relatively compact tripled Nd:YAG laser. For narrowband 354.7 nm laser radiation the vibrational Raman line of methane is at 395.7 nm, while the oxygen Raman overtone (3089 $cm^{-1}$) is at 398.4 nm which can be rejected by the interference filter.

In our paper we present the first results of methane profiling in the lower troposphere using LILAS Raman lidar from Lille University observatory platform (Hauts-de-France region,

France). The observations demonstrate that inside the PBL, $CH_4$ mixing ratio may exceed the background concentration levels by up to a factor of 2. Enhancement of the $CH_4$ mixing ratio in weak elevated aerosol layers was also detected.

## 2. Experimental setup.

The experiments described here were performed using LILAS - multiwavelength Mie-Raman lidar from the Lille University (Veselovskii et al., 2016). The lidar is based on a tripled Nd:YAG laser with a 20 Hz repetition rate, and pulse energy of 70 mJ at 355 nm. The backscattered light is collected by a 40-cm aperture Newtonian telescope. The outputs of the detectors are recorded at 7.5 m range resolution using Licel transient recorders that incorporate

both analog and photon-counting electronics. The full geometrical overlap of the laser beam and the telescope field of view (FOV) is achieved at approximately 1000 m height using a 0.75 mrad field of view. In its usual configuration, LILAS allows detection of three elastic backscattered



signals (355, 532, 1064 nm), rotational Raman signal from $N_2$ and $O_2$ molecules at approximately 530 nm (Veselovskii et al., 2015), vibrational nitrogen and water vapor Raman

signals at 387 nm and 408 nm respectively. Aerosol backscattering and extinction coefficients at 532 nm and 355 nm are calculated from Mie-Raman lidar measurements, as described in Ansmann et al. (1992). To perform CH4 measurements shown in this paper, we modified the water vapor channel (408 nm interference filter was replaced by the methane filter centered at 395.7 nm). The dichroic mirror in the receiver did not provide efficient separation of the nitrogen

(387 nm) and the methane (395.7 nm) Raman components, hence this mirror was changed for a beam splitter, reflecting approximately 10% of 387 nm component in the nitrogen channel and transmitting about 90% of 395.7 nm signal to the methane channel. Thus operation of the Raman channel at 387 nm was degraded. The Alluxa interference filter in the methane Raman channel has the bandwidth of 0.3 nm and the peak transmission above 80%. Suppression of 355 and 532

nm radiation was specified by the manufacturer to be greater than 12 orders. For additional suppression of 355 elastic backscatter signal, the interference filter was combined with the notch filter. During initial test phase, the UV glass filter was also added to verify that no 532 nm backscatter leaks into the methane channel. The intensity of the oxygen overtone is approximately three times the intensity of the methane line (Heaps and Burris, 1996) while the

filter manufacturer specifies the suppression at 398.4 nm to be above $10^6$, hence the contribution of the oxygen overtone is negligible. The Raman methane measurements were performed in the photon counting mode and at night only.

### 3. Numerical simulation

115        Numerical simulation was performed to estimate the power of the Raman backscatter for the background methane mixing ratio. The lidar equation describing the number of detected photons $N_x^{ph}$, scattered by molecule "x" at distance z due to a single laser pulse can be written as:

$$N_x^{ph}(z) = O(z)A_x \frac{E}{\hbar\nu}\triangle z \frac{S}{z^2} N_x \sigma_x \exp\left\{-\int_0^z (\alpha_L^a + \alpha_L^m + \alpha_x^a + \alpha_x^m)dz'\right\} \qquad (1)$$

Here O(z) is the geometrical overlap factor, $A_x$ is an efficiency factor, including the transmission

of the optics and the quantum efficiency of the detectors. E and hν are the laser pulse and the photon energies, Δz - range resolution, S - receiving telescope area, $N_x$ – number concentration of molecule "x" and $\sigma_x$ is the differential Raman scattering cross section of molecule "x", α – is





the extinction coefficient, where superscripts "a" and "m" indicate aerosol and molecular contributions, respectively. Subscripts "L" and "x" correspond to the laser wavelength $\lambda_L$ and to the wavelength of Raman backscatter $\lambda_x$.

Table 1 shows the parameters of $H_2O$, $CO_2$ and $CH_4$ molecules, such as Raman frequency shift and Raman differential scattering cross section $\sigma_x$, normalized to the cross section of nitrogen $\sigma_{N_2}$. Results are presented for an excitation wavelength of 337 nm basing on Weber (1979). The table also provides typical concentrations of gases in the troposphere. The efficiency of detection of molecule "x" is determined by the factor $n_x \times \dfrac{\sigma_x}{\sigma_{N_2}}$ ($n_x$ is the molecule "x" mixing ratio), which is approximately $10^4$ for the $H_2O$ molecule and about 320 for $CO_2$. However for $CH_4$ this factor is about 20 times lower than for $CO_2$, so detection of the methane background concentrations demands a powerful Raman lidar and significant signal accumulation time.

The lidar derived mixing ratio of methane can be calculated from the ratio of $CH_4$ and $N_2$ lidar Raman signals ($P_{CH_4}$ and $P_{N_2}$), corrected for the aerosol and molecular differential extinction:

$$n_{CH4}(z) = K \frac{P_{CH_4}}{P_{N_2}} \exp\left\{-\int_0^z \left[ \alpha_{N_2}^a \left(1 - (\frac{\lambda_{CH_4}}{\lambda_{N_2}})^{-\gamma}\right) + \alpha_{N_2}^m \left(1 - (\frac{\lambda_{CH_4}}{\lambda_{N_2}})^{-4}\right) \right] dz' \right\} \qquad (2)$$

Here $\lambda_{N_2}$ and $\lambda_{CH_4}$ are the wavelengths of nitrogen and methane Raman components; $\alpha_{N_2}^a$, $\alpha_{N_2}^m$ are the aerosol and molecular extinctions at $\lambda_{N_2}$; $\gamma$ is the Ångstrom exponent and K – is the calibration constant. In our measurements, we assume that the $CH_4$ mixing ratio above the boundary layer is 2 ppm and this value was used for calibration purpose. The calibration, in principle, can be performed from first principles by using a calibration lamp with known spectrum, as it has been done for Raman water vapor lidars (Venable et al., 2011). The methane mixing ratio in (2) is calculated from the ratio of the lidar signals, so the geometrical overlap factors are at least partially compensated and thus measurements below the height of the full overlap are possible. We still need to extrapolate the extinction coefficient to the region of incomplete overlap, however the influence of the aerosol differential extinction term in (2) is lower than in the water vapor measurements due to the lower wavelength separation between nitrogen and methane Raman components.



To estimate the statistical uncertainties of methane detection, we assume that a uniform aerosol layer extends from the ground up to 2 km height. In modeling the aerosol extinction coefficients at 355 nm, extinction values of 0.05, 0.1, and 0.2 km$^{-1}$ were considered. The number of detected photons was calculated from (1) for $\Delta z$=100 m, $A_X$=0.1 and $n_{CH4}$=2 ppm. The assumed laser pulse energy was 70 mJ at 354.7 nm, which corresponds with the LILAS laser

energy during the observations reported. The nitrogen Raman scattering cross section of 5.4*10$^{-31}$ cm$^2$/sr at 488 nm is taken from (Penney et al., 1974) and recalculated for 355 nm. Finally, assuming that $\dfrac{\sigma_{CH_4}}{\sigma_{N_2}}$ = 8.2 (Weber, 1979), the value $\sigma_{CH_4}$=1.9*10$^{-29}$ cm$^2$/sr at 355 nm was used.

Statistical uncertainties of the measurements are determined mainly by the weak CH$_4$ Raman backscatter and, in the absence of background noise, the uncertainty can be estimated as

$$\varepsilon \approx \frac{1}{\sqrt{N_{CH_4}^{ph}}} .$$

Fig.1 shows vertical profiles of statistical uncertainties for three values of aerosol extinction coefficient: 0.05, 0.1, 0.2 km$^{-1}$ and a signal averaging time of 4 hours. The figure shows also the photon counting rate in the methane Raman channel $v_{CH_4} = N_{CH_4}^{ph} \dfrac{2\Delta z}{c}$, where $c$ - is the speed of the light. For the clean atmosphere ($\alpha_{355}$=0.05 km$^{-1}$) the measurements with

uncertainty below 10% are possible up to 4 km, while for $\alpha_{355}$=0.2 km$^{-1}$ the corresponding range decreases to 3 km. The simulation results confirm the necessity of long-term (several hours) signal accumulation in methane measurements using Raman lidar.

### 4. Results of measurements.

Measurements were performed at Lille University observatory platform, France, during the period May-June, 2018. In total, 20 nighttime observation sessions were accomplished. Fig.2 shows CH$_4$ and N$_2$ Raman lidar signals together with the backscattered signal at 1064 nm on the night of 14-15 June 2018. The results are averaged over the temporal interval $\tau_{av}$=4.0 hours. Aerosols are mainly located below 1700 m (maximal value of aerosol extinction $\alpha_{355}$ inside the

PBL is about 0.1 km$^{-1}$), though a weak aerosol layer is also visible in the 1064 nm lidar signal in the 2.5 – 4.0 km height interval. The photon counting rate in the methane channel at 2000 m is



about 1.8 KHz, which agrees with simulation results in Fig.1 for $\alpha_{355}$=0.1 km$^{-1}$. The profile of

the methane mixing ratio calculated from the measurements in Fig.2 and averaged over 100 m

height bins is shown in Fig.3. The same figure provides the profile of the backscattering

coefficient at 532 nm. The mixing ratio is given in arbitrary units, assuming that the value of 1.0

corresponds to $n_{CH_4}$=2 ppm. The mixing ratio inside the PBL exceeds the corresponding values

above 4000 m by approximately a factor of 2. The profiles of $n_{CH_4}$ and $\beta_{532}$ are not correlated:

inside the PBL the maximum of $\beta_{532}$ is at a height of 1500 m, while the maximum of $n_{CH_4}$ is at

1100 m. The backscattering coefficient $\beta_{532}$ of the weak aerosol layer at 3500 m is about $7.6*10^{-5}$

185  km$^{-1}$sr$^{-1}$, which is almost a factor of 50 lower than the maximum value of $\beta_{532}$ inside the PBL. In

this elevated layer, the CH$_4$ mixing ratio also increases, however $n_{CH_4}$ at 3500 m is 1.5, which is

close to the values in the PBL. Thus, the enhancement of $n_{CH_4}$ at 3500 nm is very unlikely to be

an artifact related to aerosol interference.

The derived methane profiles exhibited strong night to night variation. Fig.4 shows the

190  results of six measurement sessions, representing nights with different aerosol loading. The night

of 12-13 June (Fig.4a) was characterized by a low aerosol backscattering coefficient in the 500 –

4000 m range ($\beta_{532}$ is below $2*10^{-4}$ km$^{-1}$sr$^{-1}$) and the mixing ratio shows no significant deviation

from the 1.0 value in the whole height range. By contrast, on the night of 20-21 May (Fig.4b) a

scattering layer with peak value $\beta_{532}$=0.09 km$^{-1}$sr$^{-1}$ occurs in the 2500 – 3100 m height range.

Low lidar ratio (about 20 sr) and low depolarization ratio (below 5%) indicate that this layer is

likely a water cloud. Strong cloud scattering demonstrates no influence on the mixing ratio,

which is about 1.0 in the center of the cloud, proving that the interference filters provide

sufficient rejection of elastic scattering. It should be mentioned also that the Raman band of the

liquid water extends from 395 nm to 409 nm (Reichardt, 2014), so potentially it can be an

interfering factor in the methane measurements. However, Fig.4b does not reveal a noticeable

effect of liquid water Raman scattering on the methane profile due to the narrowband filter in the

CH$_4$ channel. Cloud layers occurred also on 26-27 May and 27-28 May (Fig.4c,d) at a height of

approximately 4000 m with maximum values of backscattering coefficients of $\beta_{532}$=0.006 km$^{-1}$sr$^{-1}$

and 0.02 km$^{-1}$sr$^{-1}$, respectively. As in Fig.4b, the presence of clouds does not influence the

methane measurements. The vertical variation of methane content was related to the PBL height,

as can be concluded from a comparison of Fig.4e and Fig.4f. On 30-31 May the aerosol is





confined below 2000 m, while on 2-3 June it is below 750 m. Respectively, the $n_{CH_4}$ decreases

from 2.4 at 500 m to 1.0 at 2000 m in the first case, while in the second case the background

level of 1.0 is observed for the heights above 750 m.

On 2-3 June, the increase of CH4 mixing ratio at 3400 m correlates with a weak aerosol

layer ($\beta_{532} < 10^{-4}$ km$^{-1}$sr$^{-1}$) at the same height. It is interesting that a stronger aerosol layer at 2300

m is not accompanied by an increase in $n_{CH_4}$. Enhancement of $n_{CH_4}$ in weak elevated aerosol

layers was observed several times during the campaign. One such observation session was on the

night 13-14 June 2018. Fig.5 shows the temporo-spatial distribution of the range corrected lidar

signal at 1064 nm and the particle depolarization ratio at 532 nm for this session. Most of the

aerosols are below 2000 m, but there is an elevated layer in the 3000 – 5000 m height range. The

depolarization ratio inside the PBL is low ($\delta_{532} < 5\%$) while in the elevated layer $\delta_{532}$ increases up

to approximately 18%. The profiles of aerosol backscattering coefficient $\beta_{532}$, particle

depolarization $\delta_{532}$ and CH4 mixing ratio on 13-14 June for the temporal interval of 22:00 –

02:00 UTC are given in Fig.6. In the PBL the mixing ratio is about 2.0, and in the elevated layer

the $n_{CH_4}$ demonstrates also an increase up to approximately 1.5.

To understand the origin of this elevated layer, a ten-days back-trajectory analysis, for the

air mass over Lille, at 4000 m, on 14 June 2018 at 00:00 UTC, was performed using the HSPLIT

model (Stein et al., 2015; Rolph et al., 2017). According to the analysis, the air mass was

transported from eastern Asia (Russia and China) to North America and then over the Atlantic

Ocean to Europe. Large-scale boreal fire activities were detected near the border of Russia and

China in early June, thus the air mass at 4000 m could have contained fire emissions. Fig.7 (a)–

(d) plot the transport pathway of the air mass overlaid with the CO columnar concentration maps

on 03, 06, 09 and 12 June, respectively. The CO concentration is derived from AIRS Level 3 CO

products (Texeira, 2013). The propagation of the air mass is clearly coincident with the transport

of CO plumes. Studies have shown that CO$_2$, CO and CH$_4$ are among the main products of

boreal forest fires (Hao et al., 1993; Kasischke et al., 2002; Worden et al., 2013). CO originating

from boreal fires is positively correlated with CH$_4$ concentration, however, the CH$_4$ product of

AIRS is not as mature as the CO product due to the low sensitivity to CH$_4$ in the lower

troposphere, so CO is a favorable tracer of fire emissions. In Fig.7 (a), intense CO plumes are

detected at the origin of the trajectory, which is close to the fire activities. Hence, it is possible



that the observed methane plume comes from fire emissions in eastern Asia. Aged smoke particles mixed with Asian dust particles could be the reason for the high particle depolarization ratio observed in the elevated layer.


### 5. Discussions and conclusion

The results presented here demonstrate the feasibility of profiling the background mixing ratios of methane in the lower troposphere using Raman lidar. The photon counting rate in the methane Raman channel agrees with numerical simulation for typical aerosol loading and a

background $CH_4$ mixing ratio of 2 ppm, which confirms that we observe the methane Raman scattering. In our measurements we always observed enhanced concentrations of the methane inside the PBL, compared to aerosol free regions, thus analysis of methane ground sources in Northern France is in our upcoming plans.

Raman measurements of $CH_4$ mixing ratio close to 2 ppm is a challenging task due to

different potential interfering factors, such as leakage of the elastic signal into the Raman channel, contribution of liquid water Raman scattering and aerosol fluorescence. Measurements performed inside the clouds revealed no interfering of elastic signal or Raman liquid water spectra. Estimation of aerosol fluorescence contributions is more difficult. The aerosol fluorescence at wavelengths above 440 nm was reported recently by Reichardt et al. (2017). For

profiling, the authors had to integrate the fluorescence signal over the spectral range of approximately 80 nm. In our system the filter bandwidth is only 0.3 nm, so we expect that the fluorescence contribution is suppressed. The $CH_4$ profiles are not always correlated with aerosol backscattering, which corroborates the hypothesis that, in the PBL, not aerosol fluorescence but methane is measured. However, we cannot completely exclude fluorescence contribution. To

measure and correct for it, if necessary, in future measurements we plan to introduce an additional control channel close to 393 nm where no strong Raman lines exist and only fluorescence can be detected (Reichardt, 2014).

One of the main problems in the measurements presented is the long signal accumulation time, which was about 4 hours in our case. A more powerful laser is needed to improve the

temporal resolution of the measurements. Today, compact diode pumped lasers, with pulse energy of 60 mJ at 355 nm and 200 Hz repetition rate have become widely available (e.g. www.quantel-laser.com/en/products/item/q-smart-dpss-650-mj.html), so it is possible to decrease



the measurement time to less than 30 minutes. Numerous lidar technologies developed previously for $H_2O$ Raman systems can be used for the methane Raman lidar. In particular, the calibration technique based on the tungsten lamp spectrum, can provide absolute values of methane mixing ratio from first principles (Venable et al., 2011).

Raman lidars for $CH_4$ monitoring cannot, of course, compete with airborne DIAL systems in sensitivity and accuracy, especially when column concentrations are considered. However, when one needs to evaluate the vertical profile of methane concentrations through the boundary layer, the Raman lidar may have some advantages. In particular, the IR DIALs can profile methane only in the region loaded with aerosol, while Raman lidar is capable to profile in the aerosol free atmosphere also. Our results demonstrate that conventional Mie-Raman lidars designed for aerosol and the water vapor observations can be relatively easy modified for methane observations.



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

| Molecule | Frequency, cm$^{-1}$ | $\dfrac{\sigma_x}{\sigma_{N_2}}$ | Typical values of $n_x$, ppm | $n_x \times \dfrac{\sigma_x}{\sigma_{N_2}}$, ppm |
|---|---|---|---|---|
| $H_2O$ | 3657 | 3.1 | $3*10^3$ | $\sim 10^4$ |
| $CO_2$ | 1285 | 0.8 | 400 | 320 |
| $CH_4$ | 2914 | 8.2 | 2 | 16.4 |





**Figures**

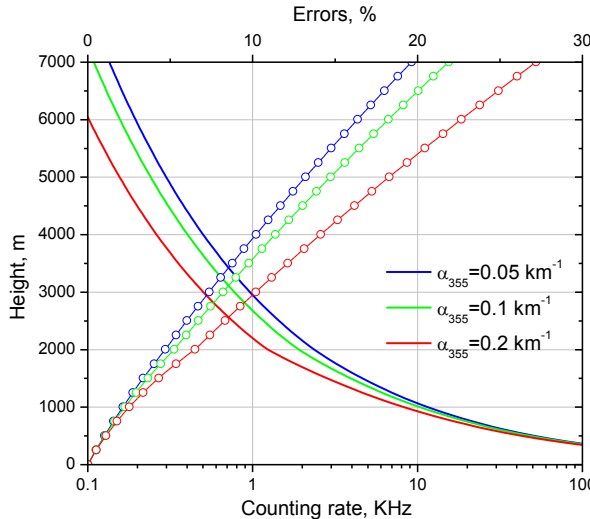

Fig.1. Modeled photon counting rate (lines) and statistical uncertainties of the methane mixing ratio measurements (lines + symbols) for a 2 ppm methane concentration and three values of aerosol extinction coefficient $\alpha_{355}$=0.05, 0.1, 0.2 km$^{-1}$. Aerosol extends from z=0 to z=2000 m.

Signal averaging time is 4 hours.





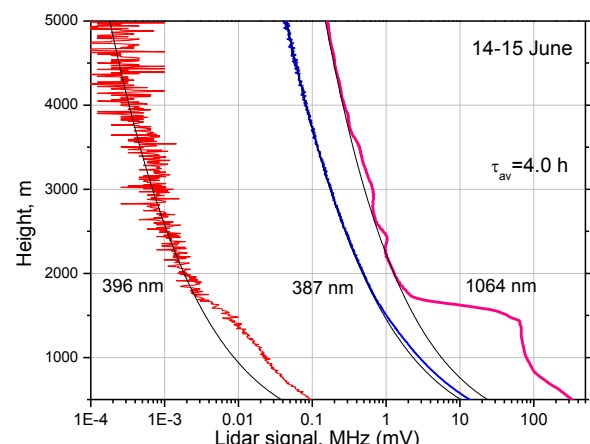

Fig.2. Lidar signals corresponding to elastic scattering at 1064 nm, nitrogen Raman scattering at 387 nm and methane Raman scattering at 396 nm on the night 14-15 June 2018. The units are MHz for 387 and 396 nm, and mV for 1064 nm. Black lines show the profiles of molecular scattering. Measurements were performed from 22:00 to 02:00 UTC, signals averaging time $t_{aver}$=4.0 hours.






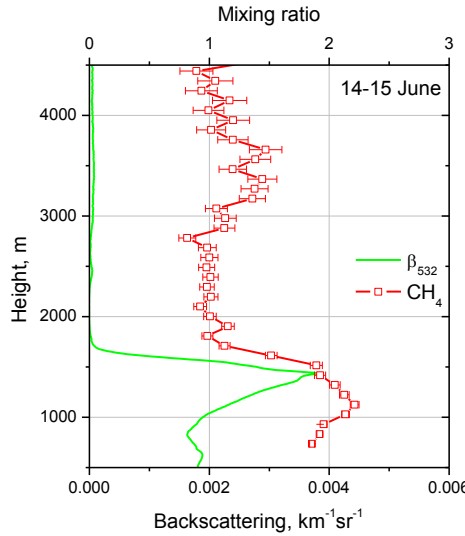

Fig.3. Vertical profiles of aerosol backscattering coefficient at 532 nm and methane mixing ratio calculated from measurements on 14-15 June for the same temporal interval as in Fig.1. Mixing ratio is uncalibrated and the value 1.0 corresponds to approximately 2 ppm.






Fig.4. Vertical profiles of the methane mixing ratio and aerosol backscattering at 532 nm for six night measurement sessions: (a) 12-13 June, (b) 20-21 May, (c) 26-27 May, (d) 27-28 May, (e) 30-31 May, (f) 2-3 June 2018. Mixing ratios are not calibrated and the value 1.0 corresponds to approximately 2 ppm. Signals averaging time $\tau_{av}$ is given in hours.





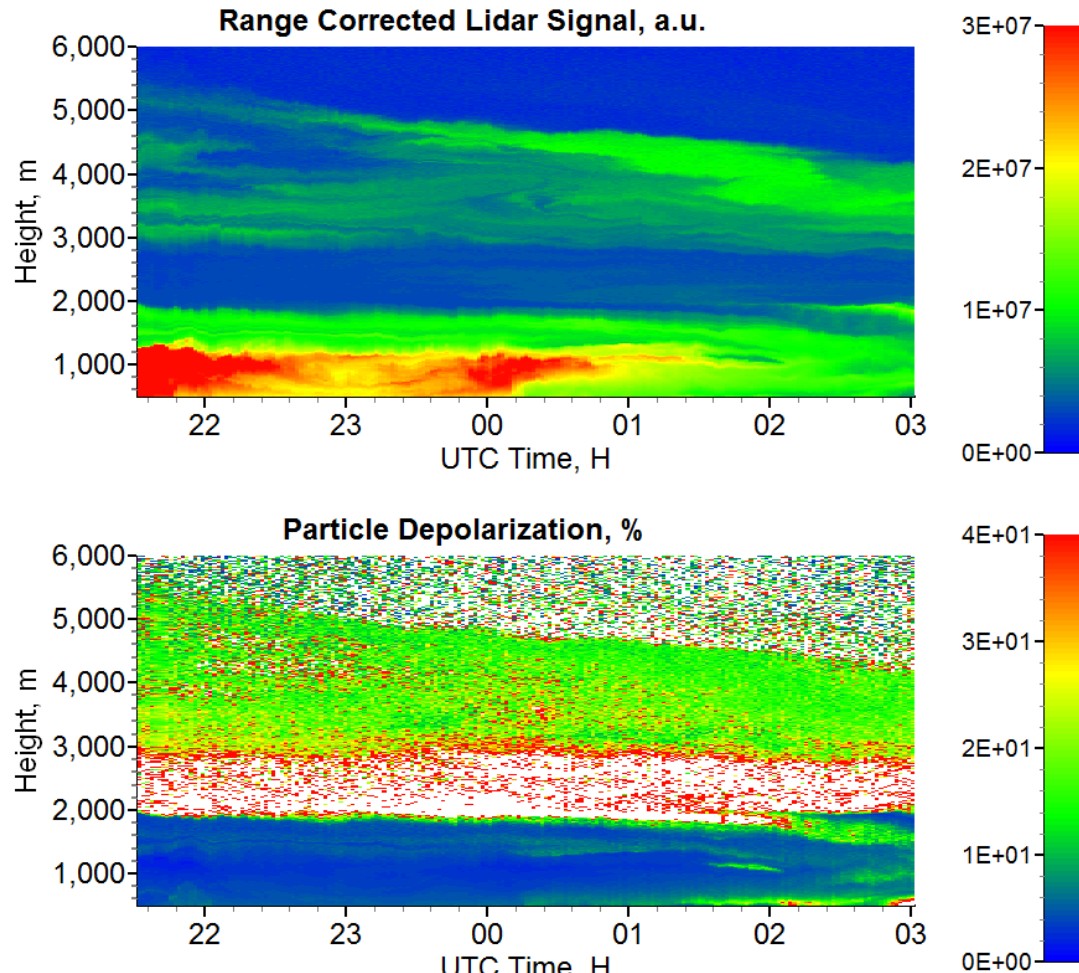


Fig.5. Range corrected lidar signal at 1064 nm and the particle depolarization ratio at 532 nm for

the night 13-14 June 2018.






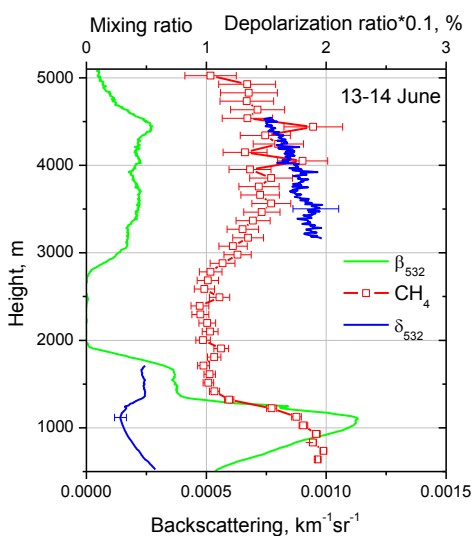


Fig.6. Aerosol backscattering coefficient $\beta_{532}$, particle depolarization $\delta_{532}$ and $CH_4$ mixing ratio on 13-14 June 2018 for temporal interval 22:00 – 02:00 UTC. Values of $\delta_{532}$ are multiplied by factor 0.1. Mixing ratios are not calibrated and the value 1.0 corresponds to approximately 2 ppm.




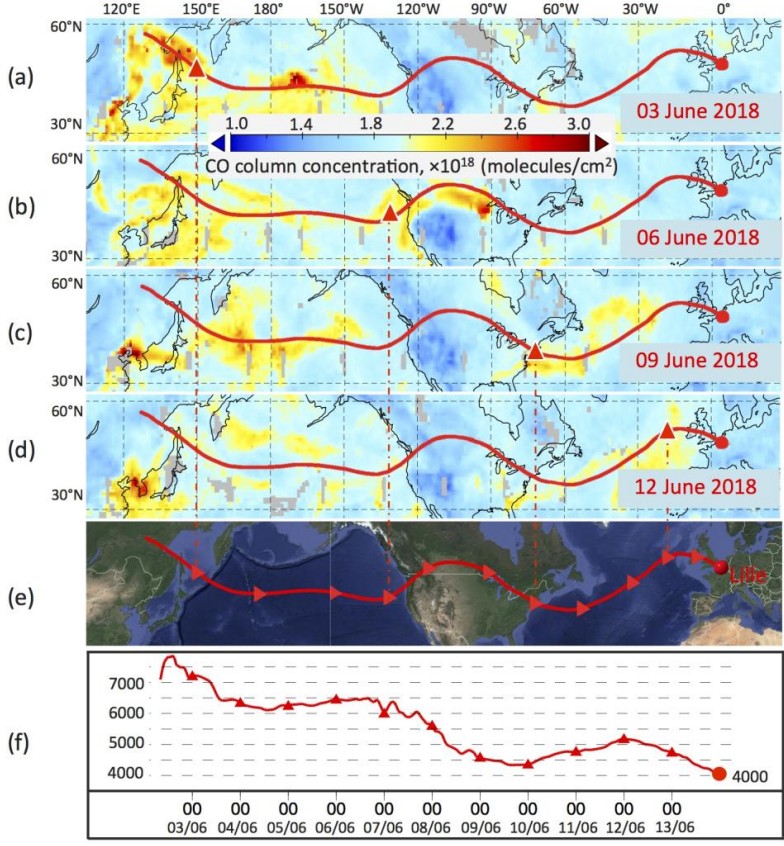

Fig.7. Ten-day backward trajectories for the air mass in Lille at altitude 4000 m on 14 June 2018

at 00:00 UTC. Plots (a)—(d) show the trajectory pathways overlaid with CO columnar concentration maps retrieved from AIRS data. The triangles represent the location of the traced air mass on corresponding dates. Plots (e) and (f) show the trajectory and vertical propagation of air mass.
