# Peer review of "Profiling of CH4 background mixing ratio in the lower troposphere with Raman lidar: a feasibility experiment."

_Atmospheric Measurement Techniques, 2018_

## Referee Comment (RC1) · S. Bobrovnikov (Referee) · 16 Oct 2018

Review of the article: Profiling of CH4 background mixing ratio in the lower troposphere with Raman lidar: a feasibility experiment. Igor Veselovskii, Philippe Goloub, Qiaoyun Hu, Thierry Podvin, David N. Whiteman, Michael Korenskiy, and Eduardo Landulfo.

Article of Veselovsky at All. "Profiling of the CH4 background mixing ratio in the lower troposphere with Raman lidar: a feasibility experiment" is devoted to the problem of remote control of small gas components of the atmosphere. At the same time, the problem of monitoring the total content and spatial distribution of the concentration of greenhouse gases is of particular practical importance in terms of predicting climate change. Methane is one of the most important climate-forming components of the atmosphere, and therefore there is an obvious need for the development of new methods and means of monitoring methane content in the atmosphere. The authors provide a fairly comprehensive analytical overview of the achievements in the field of remote monitoring of methane content in the atmosphere and substantiate the prospects for applying the Raman effect to create a lidar technology for remote monitoring of the vertical distribution of methane concentration.

Taking into account that we are talking about background concentrations of methane (about 2 ppm), experimenters understand that the determination of such low concentrations using the Raman effect is a very difficult task. However, as we see, the difficulties do not stop the authors and as convincingly shown in the publication, this problem can be successfully solved.

Obviously, the authors of the publication are experienced experimenters, lidar developers, who understand well the problems of applying the Raman effect and are able to find the right technical solutions. This is evident from the fact how carefully the authors relate to solving the problem of spectral suppression of possible interfering factors, such as un shifted scattering laser line, an overtone of oxygen, which the experimenters do not always remember, Raman on liquid water in the clouds. At the same time, the samples of interference filters made by the newest technology and having unique characteristics are correctly selected.

It also seems appropriate to carry out preliminary mathematical modeling in order to estimate the expected magnitude of lidar responses, and determine the accumulation time of signals to achieve a given accuracy of measuring the mixing ratio. And the fact that in real experiments the magnitude of lidar responses in the methane channel corresponds to the calculated values is an indirect confirmation of the correctness of the chosen technical solutions.

It should be noted that the most difficult task of lidar measurements of background concentrations of greenhouse gases using the Raman effect in the case of methane is somewhat simplified due to the extremely large scattering cross section of the methane molecule ($1.9 * 10^{-29}$ cm$^2$/sr), which is more than 8 times larger than the cross section of the nitrogen molecule. However, even in this case, experimenters are required to display not a small amount of mastery in constructing equipment and taking measurements.

As shown in the publication, the results of lidar measurements of the vertical course of the methane mixing ratio do not contradict generally accepted ideas about the spatial distribution of methane in the atmospheric boundary layer. The absence of a direct correlation between the aerosol backscatter coefficient and the methane mixing ratio indicates a sufficient level of suppression of the unshifted scattering noise and possible liquid water Raman signal.

However, as the authors rightly point out in conclusion, it is impossible to completely exclude the fluorescence factor of an aerosol, given the high sensitivity of the lidar at the level of ppm units. The authors rightly say that the Raman lidar for sensing of water vapor mixing ratio can be easily converted into a lidar for methane sensing. However, the question of the

contribution of aerosol fluorescence to the Raman signal of methane remains open. All the same, to ensure correct measurements of the methane mixing ratio, it is necessary to build a special lidar, equipped with a more powerful laser and a special channel to control the level of aerosol fluorescence. The authors understand this and plan to carry out such work in the future. As for my opinion, I consider it more reliable to carry out research using a multichannel spectrometer, which makes it possible to see the spectral image of the lidar response and to interpret various spectral components of the signal.

Nevertheless, the presented publication convincingly proves the possibility of lidar measurements of the vertical distribution of the methane mixing ratio using lidar. It is obvious that, in the main, the presented results reflect the real vertical distribution of methane concentration. However, as long as there is no control over the level of aerosol fluorescence, it is difficult to trust the measurements of the methane mixing ratio inside the aerosol layers. The magnificent hypothesis of methane brought in the composition of the products of combustion of forest fires (Fig. 6) can also be explained as a result of the fluorescence of smoke aerosol and products of combustion or sublimation of the organic components of wood.

Summarizing, it should be stated:

1. The paper addresses relevant scientific questions within the scope of AMT.
2. The paper presents novel concept of Raman lidar technique, new tool for monitoring of methane mixing ratio, data of methane vertical distribution.
3. Substantial conclusions are mostly reached.
4. The scientific methods and assumptions are valid and clearly outlined.
5. The results are sufficient to support the interpretations and conclusions.
6. The description of experiments and calculations is sufficiently complete and precise to allow their reproduction by fellow scientists.
7. The authors give proper credit to related work and clearly indicate their own original contribution.
8. The title clearly reflects the contents of the paper.
9. The abstract provide a concise and complete summary.
10. The overall presentation is well structured and clear.
11. The language is fluent and precise.
12. Mathematical formulae, symbols, abbreviations, and units are defined and used correctly.
13. Text, formulae, figures and tables are satisfied demands for scientific publications.
14. The number and quality of references are quite appropriate.
15. The amount and quality of supplementary material is appropriate.

I believe that the article under discussion can be published in AMT without significant modifications and changes.

Head of the Center of Laser Sounding of the Atmosphere
 of the Institute of Atmospheric Optics,
Siberian Branch of the Russian Academy of Sciences,
 Professor                                                    Sergei Bobrovnikov

---

## Referee Comment (RC2) · Anonymous Referee #2 · 9 Nov 2018

General remark

Dr. Sergey Bobrovnikov (see his review in the discussion section) already wrote an almost exhausting review. Because Dr. Bobrovnikov is a pioneer in the field of Raman lidar applications and observations of trace gases and temperature (since the late 1970ies) I have only minor points to add.

The paper is well written and successfully combines experimental work with modelling results. The shown cases studies are promising and convincing.

The paper can be regarded as a highlight of AMT.

[Figure]

Minor revisions may further improve the paper.

Detailed points:

Abstract: would be nice to mention the wavelengths 395.4 nm (CH4) and the reference channel (N2, 387 nm) already in the abstract.

P2, L36: Ansmann 1998, I did not find...

P3, L77: One should mention, ...somewhere in the introduction...., the MERLIN project (ESA's mission on spaceborne methane lidar observations, with DIAL, but column-integrated...) to corroborate how important methane observations are. ESA has nice handbooks with nice introductories.

One could then mention that such CH4 Raman lidar observations in Lille could be used for ground truth activities. The launch of MERLIN is planned for 2024.

Result section:

P6, L175-176: Please do some HYSPLIT computations, provide information about the source region of the detected layers.

P7, L185-188: Again provide some information about the origin of the air masses detected.

P7. L195: Note that the apparent lidar ratio in water clouds should be 10-15sr (instead of 18.2sr) because of multiple scattering effects.

P8, L217-218: Again, information on the origin of the found air masses would be help-ful.

Enhanced depolarization can be caused by dust and by dry smoke. Are radiosonde RH profiles available. Smoke may become nonsphercial when RH is below 30%.

P9, L239: Again, would be nice to have some RH information.

P10: At the end, mention again the MERLIN mission, and that ground-based Raman

lidars are good for ground truth activities.

Figs. 3 and 4: There are layers, and the reader wants to know: what is the source?

Fig 4. Why not a temporal order: b,c,d,e,f,a?

Fig 5: Are RH profiles available (radiosonde)? Is the lofted layer dry (...then non-spherical particles) or wet (more spherical particles)? Further point: Origin of the lofted layer...?

Fig. 6: Depolarization ratios of 15-18%! Is that caused by dust or by dry smoke particles? Origin of the aerosol ....

Fig. 7: Maybe the smoke was picked up in North America ?

All in all: A nice paper!
* * *

---

## Short Comment (SC1) · 14 Nov 2018

Comment on the manuscript "Profiling of CH4 background mixing ratio in the lower troposphere with Raman lidar: a feasibility experiment" by Igor Veselovskii, Philippe Goloub, Qiaoyun Hu, Thierry Podvin, David N. Whiteman, Michael Korenskiy, and Eduardo Landulfo.

The manuscript presents a Raman lidar for remote measurements of methane. The approach of the authors is generally valid. The authors are renowned experts in the development of Raman lidar systems, as well as in the simulation of the performance of such systems and the analysis of the collected data.

[Figure]

Nevertheless, there are several points need to be addressed in more detail:

Reproducibility of the results: Over the last decade I have looked for a methane Raman signal at 2914 cm-1 with three multi-channel spectroscopic lidars: at Tsukuba, Japan (Sugimoto at all. 2012), Gwangju, Korea, and now at Hatfield, United Kingdom. During the work with all those instruments I have never managed to detect methane background signals as shown in the manuscript when using a laser power for the emitted light comparable to the one available to the authors. The multi-channel lidars I have worked with are all based on spectrometric and long-pass edge filter isolation of Raman lines rather than single bandpass interference and notch filters as used by the authors. The system in Japan used 100mJ@355nm at 30Hz repetition rate and a 100 cm telescope. At Gwangju we used about 200mJ at 10Hz and a 40 cm telescope. We are not able to observe the background methane signal even with a laser energy of about 300mJ at 10Hz (40 cm telescope) in the spectrometric lidar system at Hatfield. With all these systems we can observe nitrogen and H2O Raman signals with counting rates of tens or even hundreds MHz when using emission energy below 200mJ, but nothing above the noise levels in the 396nm channel.

Signal isolation: The filter the authors use to isolate the methane line should be described in more details. In fact, the Alluxa interference filter (395.7-0.3 OD12 Ultra Narrow Bandpass Filter) has a rejection ratio (optical depth, OD) of 12 only for some wavelengths. According to the manufacturer's web page (https://www.alluxa.com/optical-filter-catalog/ultra-narrow-bandpass/395-7-0-3-od12-ultra-narrow-bandpass.html) this filter has "Blocking Range(s) OD12 (By Design): 353 to 389 nm, 403 to 443 nm, 485 to 540 nm; OD5: 300 to 353 nm, 443 to 485 nm, 540 to 1100 nm". The filter has OD5 for some of the pure-rotational anti-Stokes lines around 352nm. Using an additional notch filter can provide a good suppression of the pure rotational Raman signal. However, the optical depth is OD5 for almost all anti-Stokes Raman spectra (351 nm to 309 nm) including anti-Stocks scattering by nitrogen, oxygen, and H2O molecules. The optical depth is OD5 for wavelengths larger than 540nm which includes the Raman signals by

nitrogen, oxygen and H2O when using a laser at 532nm. The authors should provide the curves of ATR (Attenuation-Transmission and Reflection) of the particular filter and discuss the suppression/rejection ratio for pumping of the anti-Stokes band by 355nm as well as excitation of the Stokes band of Raman spectra pumped at 532nm.

Interference with other signals: Although the methane Raman line appears to be well isolated there are other Raman lines of atmospheric compounds very close to it. Namely: Propane line at 2890cm-1 (395.3nm) - 0.4nm shift from the author's Interference filter peak transmission, Ethanol 2943cm-1 (396.1nm) - 0.4nm shift from the author's Interference filter peak transmission with Raman cross section of the same magnitude as methane, Methanol 2v6 line at 2955cm-1 (396.3nm) - 0.6nm shift from the Interference filter peak transmission which can be connected to the methane cycle and its interaction with water vapour.

Lack of reference data: As the study assesses the feasibility for profiling of methane with a modified multiwavelength Raman lidar, the authors have to compare their findings to independent data (especially regarding the background concentration) from in-situ measurements or DIAL observations. The conclusion made on page 1, line 20 "The measured methane profiles do not correlate with aerosol backscattering, which corroborates the hypothesis that, in the PBL, not aerosol fluorescence but CH4 is observed." is no proof of concept unless supported by independent observations. In the present form, the reliability of the results is highly speculative.

Measurement setup: Additional information regarding detector sensitivity and parameters (high voltage and discriminator levels) should be provided. If we take a background concentration of 2ppm and the ratio of Raman backscatter cross sections (methane to nitrogen) of 8, the nitrogen signal should be approximately 1.6x10ˆ5 (160000) higher than the methane signal. However, Fig. 2 shows a nitrogen signal (378nm) that is only approximately 100 times higher than the methane (396nm) signal. If one accounts for the 10% beam splitter applied for the 378nm channel, then the nitrogen signal seems to be 1000 times higher than the methane signal. Hence, the counting rates in the 396-

nm channel appear to be about two orders of magnitude higher than expected. The authors need to provide information on how they managed to obtain that high count rate.

Data consistency: The authors assume that the background methane concentration in the free troposphere is 2ppm and can differ in the PBL "inside the planetary boundary layer (PBL)" (Page 2, line 47). Yet they present enhanced methane mixing ratios at altitudes far above the PBL. In Fig. 6, the methane mixing ratio is significantly higher from 3000m to 5000m. The authors should discuss possible mechanisms that could lead to the formation of methane plumes in these height ranges and/ or persist in the free troposphere.

Fluorescence: Sugimoto et al. (2012) show that fluorescence can be observed in case of pumping at 355nm. Though the fluorescence maximum has been observed at about 460-470nm, fluorescence interference should be considered as an interfering factor in measurements for which optical pumping at 355nm is used.

Reference: Nobuo Sugimoto, Zhongwei Huang, Tomoaki Nishizawa, Ichiro Matsui, and Boyan Tatarov, "Fluorescence from atmospheric aerosols observed with a multi-channel lidar spectrometer," Opt. Express 20, 20800-20807 (2012)

---

## Author Comment (AC1) · 16 Nov 2018

We are grateful to Boyan Tatarov for reading the manuscript and providing important suggestions. Raman measurements at ppm level are challenging and many factors should be considered.

*"Reproducibility of the results: Over the last decade I have looked for a methane Raman signal at 2914 cm-1 with three multi-channel spectroscopic lidars: at Tsukuba, Japan (Sugimoto at all. 2012), Gwangju, Korea, and now at Hatfield, United Kingdom. During the work with all those instruments I have never managed to detect methane background signals as shown in the manuscript when using a laser power for the emitted light comparable to the one available to the authors. The multi-channel lidars I have worked with are all based on spectrometric and long-pass edge filter isolation of Raman lines rather than single bandpass interference and notch filters as used by the authors. The system in Japan used 100mJ@355nm at 30Hz repetition rate and a 100 cm telescope. At Gwangju we used about 200mJ at 10Hz and a 40 cm telescope. We are not able to observe the background methane signal even with a laser energy of about 300mJ at 10Hz (40 cm telescope) in the spectrometric lidar system at Hatfield. With all these systems we can observe nitrogen and H2O Raman signals with counting rates of tens or even hundreds MHz when using emission energy below 200mJ, but nothing above the noise levels in the 396nm channel."*

We are not the only ones who see the Raman signal from background methane, for example (Heaps, Wm. S. and Burris 1996). Keeping in mind that cross section of methane is about 8 times higher than that of nitrogen, for 2ppm mixing ratio the methane signal should be $10^6/16=6*10^4$ weaker than that of N2. For counting rate 100 MHz in nitrogen channel, methane signal should be ~1.7 KHz, which is low but definitely detectable.

*"Measurement setup: Additional information regarding detector sensitivity and parameters (high voltage and discriminator levels) should be provided. If we take a background concentration of 2ppm and the ratio of Raman backscatter cross sections (methane to nitrogen) of 8, the nitrogen signal should be approximately 1.6x10^5 (160000) higher*

Actually $6*10^4$

*than the methane signal. However, Fig. 2 shows a nitrogen signal (378nm) that is only approximately 100 times higher than the methane (396nm) signal. If one accounts for the 10% beam splitter applied for the 378nm channel, then the nitrogen signal seems to be 1000 times higher than the methane signal. Hence, the counting rates in the 396- nm channel appear to be about two orders of magnitude higher than expected. The authors need to provide information on how they managed to obtain that high count rate."*

Yes, it is important to compare nitrogen and methane Raman signals. However lidar signals in Fig.2 are not calibrated and can't be used for this: only small portion of spitted nitrogen Raman backscatter was used for the measurements. To perform calibration in Fig.1 we show nitrogen Raman signal before system modification (normal operation) and after. Before modification we glued analog and photon counting signals so equivalent counting rate at 1000 m height is about 500 MHz. Comparing the signals we estimate attenuation in modified N2 channel to be about factor 185. Methane signal is also shown in Fig.1 and  at 2000 m it is about $5*10^4$ lower than

nitrogen signal. Transmission of N2 interference filter was slightly lower than that in CH4 channel, so we find the agreement between nitrogen and methane signals to be very reasonable.

[Figure]

Fig.1.
All these estimations are added to the revised manuscript.

*"Signal isolation: The filter the authors use to isolate the methane line should be described in more details. In fact, the Alluxa interference filter (395.7-0.3 OD12 Ultra Narrow Bandpass Filter) has a rejection ratio (optical depth, OD) of 12 only for some wavelengths. According to the manufacturer's web page (https://www.alluxa.com/opticalfilter-catalog/ultra-narrow-bandpass/395-7-0-3-od12-ultra-narrow-bandpass.html) this filter has "Blocking Range(s) OD12 (By Design): 353 to 389 nm, 403 to 443 nm, 485 to 540 nm; OD5: 300 to 353 nm, 443 to 485 nm, 540 to 1100 nm.*
*The filter has OD5 for some of the pure-rotational anti-Stokes lines around 352nm. Using an additional notch filter can provide a good suppression of the pure rotational Raman signal. However, the optical depth is OD5 for almost all anti-Stokes Raman spectra (351 nm to 309 nm) including anti-Stocks scattering by nitrogen, oxygen, and H2O molecules. The optical depth is OD5 for wavelengths larger than 540nm nitrogen, oxygen and H2O when using a laser at 532nm. The authors should provide the curves of ATR (Attenuation-Transmission and Reflection) of the particular filter and discuss the suppression/rejection ratio for pumping of the anti-Stokes band by 355nm as well as excitation of the Stokes band of Raman spectra pumped at 532nm."*

The filter  transmission curve, simulated by Alluxa, is shown in Fig.2.

[Figure]

Fig.2. Transmission curve of the methane filter (simulation).

Filter transmission at 395.7 is about 80%, so suppression of rotational lines near 350 nm was more than OD9. Additional notch filter from Edmund ( 355nm TECHSPEC® OD>4 Notch Filter), provided OD4 suppression of rotational anti-Stokes line down to 348 nm. Dichroic mirror at the entrance of the lidar receiving module removed 90% of UV radiation in 340-360 nm range So total suppression of rotational Raman signal was more than OD13.
Suppression of signals in 300-340 nm range by filter is above OD6, transmission of optical path in lidar receiving module for this wavelength range was less than 5%, so total suppression of OD7 should be sufficient to remove anti-stokes Raman scattering of nitrogen and oxygen.
To check the influence of 532nm scattering and corresponding Raman components, during test measurements we inserted an additional UV filter with transmission of less than 5% in 500 – 750 nm range. No noticeable changes in CH4 Raman signals were discovered. Moreover, during filter testing, we inserted the Glan prism in laser beam to remove 532 nm radiation. Again, we didn't see noticeable changes in CH4 signal. So we don't expect that 532 nm radiation scattering can be an issue. But definitely, in our future measurements we will try to improve transmission characteristics of CH4 channel by using additional filters. We added the filter transmission curve to revised manuscript.

*"Interference with other signals: Although the methane Raman line appears to be well isolated there are other Raman lines of atmospheric compounds very close to it. Namely: Propane line at 2890cm-1 (395.3nm) - 0.4nm shift from the author's Interference filter peak transmission, Ethanol 2943cm-1 (396.1nm) - 0.4nm shift from the author's Interference filter peak transmission with Raman cross section of the same magnitude as methane, Methanol 2v6 line at 2955cm-1 (396.3nm) - 0.6nm shift from the Interference filter peak transmission which can be connected to the methane cycle and its interaction with water vapour."*

In some special cases (extreme industrial emissions?) such interference probably may occur, but we don't expect it in free troposphere. Besides shift of ~0.4 nm significantly reduces interfering signals.

*"Lack of reference data: As the study assesses the feasibility for profiling of methane with a modified multiwavelength Raman lidar, the authors have to compare their findings to independent data (especially regarding the background concentration) from in-situ measurements or DIAL observations. The conclusion made on page 1, line 20 "The measured methane profiles do not correlate with aerosol backscattering, which corroborates the hypothesis that, in the PBL, not aerosol fluorescence but CH4 is observed." is no proof of concept unless supported by independent observations. In the present form, the reliability of the results is highly speculative."*

We agree with Boyan, that independent reference data are critically important. Unfortunately it is not easy to get vertical distribution of CH4. We plan to use Fourier spectrometer measurements in future for validation.

*"Data consistency: The authors assume that the background methane concentration in the free troposphere is 2ppm and can differ in the PBL "inside the planetary boundary layer (PBL)" (Page 2, line 47). Yet they present enhanced methane mixing ratios at altitudes far above the PBL. In Fig. 6, the methane mixing ratio is significantly higher from 3000m to 5000m. The authors should discuss possible mechanisms that could lead to the formation of methane plumes in these height ranges and/ or persist in the free troposphere."*

We discussed in the manuscript, that methane could be generated during the forest fires. On another hand, at a moment we can't completely exclude possibility the interference of fluorescence of aerosol or gases.

*"Fluorescence: Sugimoto et al. (2012) show that fluorescence can be observed in case of pumping at 355nm. Though the fluorescence maximum has been observed at about 460-470nm, fluorescence interference should be considered as an interfering factor in measurements for which optical pumping at 355nm is used.*
*Reference: Nobuo Sugimoto, Zhongwei Huang, Tomoaki Nishizawa, Ichiro Matsui, and Boyan Tatarov, "Fluorescence from atmospheric aerosols observed with a multichannel lidar spectrometer," Opt. Express 20, 20800-20807 (2012)"*

Yes, as mentioned, in future measurements we plan to introduce the control channel at 393 nm to monitor possible fluorescence. The reference is added to revised manuscript.

---

## Author Comment (AC2) · 5 Dec 2018

We are very grateful to Sergei Bobrovnikov for his detailed and positive review. In the revised manuscript we added comments regarding suppression of interfering signals and filter characteristics.

---

## Author Comment (AC3) · 5 Dec 2018

First of all, we are very grateful to Referee#2 for his careful reading of manuscript and numerous useful suggestions. Below, we respond his comments:

*"Abstract: would be nice to mention the wavelengths 395.4 nm (CH4) and the reference channel (N2, 387 nm) already in the abstract."*

Done

*"P2, L36: Ansmann 1998, I did not find:":*

Corrected

*"P3, L77: One should mention, ...somewhere in the introduction...., the MERLIN project (ESA's mission on spaceborne methane lidar observations, with DIAL, but column integrated) to corroborate how important methane observations are. ESA has nice handbooks with nice introductories. One could then mention that such CH4 Raman lidar observations in Lille could be used for ground truth activities. The launch of MERLIN is planned for 2024."*

We have added information about MERLIN mission on page 3.

*"P6, L175-176: Please do some HYSPLIT computations, provide information about the source region of the detected layers."*

Air masses in the layers were transported over the Atlantic from Canada. We have added information obtained from HYSPLIT in discussion of observed methane profiles

*"P7, L185-188: Again provide some information about the origin of the air masses detected."*

Added

*"P7. L195: Note that the apparent lidar ratio in water clouds should be 10-15sr (instead of 18.2sr) because of multiple scattering effects."*

Yes, it is correct. We changed for "below 20 sr".

*"P8, L217-218: Again, information on the origin of the found air masses would be helpful. Enhanced depolarization can be caused by dust and by dry smoke. Are radiosonde RH profiles available. Smoke may become nonsphercial when RH is below 30%."*
*"Fig 5: Are RH profiles available (radiosonde)? Is the lofted layer dry :then nonspherical particles) or wet (more spherical particles)? Further point: Origin of the lofted layer...?"*

We had radiosonde data from Trappes (48.77_N, 1.99_E, Paris, France) and Beauvechain (50.78_N, 4.76_E, Essen, Belgium), corresponding plot on 14 June at 00:00 is attached.

[Figure]

Paris data show RH to be approximately 40% above 4000 m, Essen measurements provide values below 30% at 4000 m. So yes, humidity is not high and smoke, in principle, can depolarize radiation strongly. Unfortunately extinction coefficient of the elevated layer is quite low (below 0.02 km-1), so we are not able to provide reliable lidar ratio in this layer to identify the aerosol type. It can be smoke or smoke mixed with dust as well.

*"P10: At the end, mention again the MERLIN mission, and that ground-based Raman lidars are good for ground truth activities."*

Added

"Figs. 3 and 4: There are layers, and the reader wants to know: what is the source?"

Back trajectories for both layers are similar: aerosol from Canada is transported over the Atlantic. Unfortunately, basing on HYSPLIT model we are not able to make conclusion about difference in  nature of these two close layers.

*"Fig 4. Why not a temporal order: b,c,d,e,f,a?"*

In revised manuscript we have rearranged plots in temporal order

 *"Fig. 6: Depolarization ratios of 15-18%! Is that caused by dust or by dry smoke particles? origin of the aerosol ...."*

Unfortunately we don't have enough information for ultimate conclusion. Still, usually smoke in lower troposphere doesn't present such high depolarization at 532 nm. So long transported dust mixed with smoke looks more probable.

*"Fig. 7: Maybe the smoke was picked up in North America?"*

Yes, it is possible. However depolarization at 532 nm is high, this is why  we think, that Asian origin of the particles is more probable.